



# Source Apportionment of Urban Particulate Matter using Hourly Resolved Trace Metals, Organics, and Inorganic Aerosol Components

Cheol-Heon Jeong[1], Jon M. Wang[1], Greg J. Evans[1]

[1]Southern Ontario Centre for Atmospheric Aerosol Research, University of Toronto, Toronto, M5S 3E5, Canada

*Correspondence to*: Cheol-Heon Jeong (ch.jeong@utoronto.ca) and Greg J. Evans (greg.evans@utoronto.ca)

**Abstract.** Source apportionment analysis of hourly resolved particulate matter (PM) speciation data was performed using positive matrix factorization (PMF). The data were measured at an urban site in downtown Toronto, Canada during two campaign periods (April-July, 2013; November, 2013-February, 2014), and included trace metals, black carbon, and mass spectra for organic and inorganic species (PMF$_{Full}$). The chemical composition was measured by collocated high time resolution instrumentation, including an Aerosol Chemical Speciation Monitor, an Xact metals monitor, and a seven-wavelength Aethalometer. Separate PMF analyses were conducted using the trace metal only data (PMF$_{metal}$) and organic mass spectra only (PMF$_{org}$), and compared with the PMF$_{Full}$ results. Comparison of these three PMF analyses demonstrated that the full analysis offered many advantages in the apportionment of local and regional sources compared to using the organic or metals data individually. In combining the high time resolution data, this analysis enabled i) the quantification of metal-rich sources of PM$_{2.5}$ (PM < 2.5 μm), ii) the resolution of more robust factor profiles and contributions, and iii) the identification of additional organic aerosol sources.

Nine factors were identified through the PMF$_{Full}$ analysis: five local factors (i.e. Road Dust, Primary Vehicle Emissions, Tire Wear, Cooking, and Industrial Sector) and four regional factors (i.e. Biomass Burning, Oxidised Organics, Sulphate and Oxidised Organics, and Nitrate and Oxidised Organics). The majority of the metal emissions (83%) and almost half of the black carbon (49%) were associated with the three traffic-related factors which, on average, contributed a minority (17%) of the overall PM$_{2.5}$ mass. Strong seasonal patterns were observed for the traffic-related emissions: higher contributions of resuspended road dust in spring vs. a winter high for tire wear related emissions. Biomass Burning contributed the majority of the PM$_{2.5}$ mass (52%) in June and July due to a major forest fire event. Much of this mass was due to photochemical aging of the biomass burning aerosol. On average, industrially related factors contributed almost half (49%) of the PM$_{2.5}$; most of this mass was secondary aerosol species. Nitrate coupled with highly oxidised organics was the largest contributor, accounting for 30% of PM$_{2.5}$ on average, with higher levels in winter and at night. Including the temporal variabilities of inorganic ions and trace metals in the PMF$_{Full}$ analysis provided additional structure to subdivide the low volatility oxidised organic aerosol into three sources. Resuspended road dust was identified as a potential source of aged organic aerosol.

The novelty of this study is the application of PMF receptor modeling to hourly resolved trace metals in conjunction with organic mass spectra, inorganic species, and black carbon for different seasons, and the comparison of separate PMF analyses applied to metals or organics alone. The inclusion of these different types of hourly data allowed more robust apportionment of PM sources, as compared to analysing organic or metals data individually.





## 1    Introduction

Particulate matter (PM) has been shown to have negative impacts on human health, atmospheric visibility, and radiative forcing (e.g. IPCC, 2013; HEI, 2013). The source identification and quantification of particulate matter (PM) is crucial to understanding

aerosol chemical processes and developing effective PM abatement strategies. Receptor modeling provides a method to distinguish the relative contributions of PM sources based upon measurements at receptor sites. Positive Matrix Factorization (PMF), a bilinear multivariate receptor model, is widely used to identify sources of PM in the atmosphere and provide the contribution of each source. Typically, receptor modeling of 24-hr integrated PM chemical speciation data has been used for understanding the relative contributions of different sources and providing an overview of long-term temporal or spatial patterns

of major source categories (e.g. Xie et al., 1999; Lee et al., 2003; Chen et al., 2011; Jeong et al., 2013).

Receptor modeling of high time resolution data (hourly or sub-hourly) can be used to better resolve and understand processes and sources with more rapid temporality, and thereby obtain more accurate source apportionment results. Organic aerosol (OA), a major component of PM, is not fully understood due to its complex composition and properties. Over the past years, source apportionment using high time resolution OA mass spectra measured by an aerosol mass spectrometer (AMS) or an

aerosol chemical speciation monitor (ACSM) has been a useful approach to reduce the dimensionality of complex organic fractions into more easily interpretable OA factors (e.g. Ulbrich et al., 2009; He et al., 2010; Ng et al., 2011; Crippa et al., 2013; Bougiatioti et al., 2014; Lee et al., 2015). Three primary factors: hydrocarbon-like organic aerosol (HOA), cooking organic aerosol (COA), and biomass burning organic aerosol (BBOA), are commonly identified in many locations. Additionally, two secondary organic factors: low-volatility oxygenated organic aerosol (LV-OOA) and semi-volatile oxygenated organic aerosol

(SV-OOA), have also been observed (e.g. Zhang et al., 2007; Jimenez et al., 2009; Ng et al., 2010). More recently, PMF was applied to combined organic and inorganic mass spectra to find additional organic source types and provide insight into the characteristics and processes of these sources (Chang et al., 2011; Sun et al., 2012; McGuire et al., 2014). McGuire et al. (2014) reported that oxygenated OA factors related to sulphate and nitrate were more effectively resolved by a full mass spectra method including inorganics. Thus, additional information (i.e. trace metals) may be useful in resolving a more complete understanding

of the sources of OA.

Typically, trace metals comprise a minor component of ambient $PM_{2.5}$ (PM smaller than 2.5 μm in aerodynamic diameter) on a mass basis. Nevertheless, these minor elements are very useful as they can act as identifying markers of $PM_{2.5}$ sources. Highly time resolved measurements of trace metal components add greater temporal variabilities to certain source apportionment analysis, which can assist in the identification of local sources such as traffic and industry. Using one- or two-hour resolution

metal data in receptor modeling analyses has proven to be useful in the identification of local aerosol sources in urban areas (Richard et al., 2011; Dall'Osto et al., 2013). These previous studies found local urban dust factors such as road dust, brake dust, and local industrial sources, which typically occur sporadically and last only a few hours at most. However, the low inherent concentrations of key marker metals can be a challenge when applying PMF to metals data.

This study examines three different PMF approaches using hourly time resolution data and compares the identification and

mass attribution of the PM sources thereby achieved. The data considered included trace metals, organic/inorganics species, and black carbon measured in an urban area during warm (April-July) and cold (December-February) months. Separate PMF analyses were executed for the i) trace metals, ii) organic mass spectra, and iii) combined metals, organics/inorganics, and black carbon (BC). A number of advantages to using this combined analysis were thereby identified in terms of i) quantification of metal-rich sources of $PM_{2.5}$ such as road dust, ii) resolution of more robust factor profiles for difficult to separate sources such as

HOA vs. COA, or OOA vs. BBOA, and iii) subdivision of sources of aged organic aerosol. The results from this study provide





additional insight into sources of fine particle pollutants that have high temporal variations and thereby support the development of more effective control strategies for ambient pollutants.

## 2    Experimental Methods

### 2.1 Measurement site and period

Measurements of the chemical speciation of PM were conducted at the laboratory of the Southern Ontario Centre for Atmospheric Aerosol Research (SOCAAR) at the University of Toronto. The monitoring site (43° 39' 32" N, 79° 23' 43" W), located in downtown Toronto, is ~15 m from a busy road that is surrounded by multi-story buildings. Measurements were made

during two campaign periods: April 5 to July 14, 2013 (warm months, spring and summer, average temperature = 16 ℃) and November 25, 2013 to February 11, 2014 (cold months, winter, average temperature = -5 ℃).

### 2.2 Trace metal measurements

At the SOCAAR monitoring site, ambient air was drawn through a main sampling inlet located 3 m above the ground that was equipped with a virtual impactor to remove coarse particles. Hourly concentrations of trace metals in $PM_{2.5}$ were measured using an Xact metals monitor (Xact 625, Cooper Environ.). In brief, the Xact instrument pulls ambient air through a section of filter tape at a flow rate of 16.7 lpm using a $PM_{2.5}$ sharp cut cyclone (BGI). The exposed filter tape spot then advances into an analysis area where the collected $PM_{2.5}$ is analyzed by energy-dispersive X-ray fluorescence (XRF) to determine metal mass

concentrations. The sequence of sampling and analysis continues automatically on an hourly basis. In this study, this metal monitor was setup to measure the following metals: As, Ba, Br, Ca, Cu, Fe, K, Mn, Ni, Pb, Se, Si, Sr, Ti, V, and Zn.

For quality assurance/quality control (QA/QC) of the metals data, the flow rate was calibrated using a standard flowmeter (Streamline Pro, Chinook Engineering) and regulated via ambient temperature and pressure measurements. Automatic internal quality control checks were performed on a daily basis. The internal energy alignments and intensity checks of a known

sample for Cr, Pb, and Cd were automatically conducted every midnight (00:00 to 00:30). All midnight data (00:30-01:00) after the internal performance check were excluded from this study due to the higher background levels associated with the shorter collecting times. In addition to the daily internal check, the stability of the Xact was ensured via measurement of an internal Palladium (Pd) source in every ambient sample analysis.

Multiple measurements of HEPA-filtered air and external metal standards were conducted to determine the instrument

detection limits (DL) and analytical uncertainties of each metal. The detection limit, analytical uncertainty, and average concentrations of trace metals are summarized in the Supplement and Table S1.

### 2.3 Organic and inorganic composition measurements

Non-refractory organic and inorganic PM compositions were measured using an ACSM (Aerodyne) at a time resolution of 30 minutes. The ACSM is capable of providing online quantitative data on mass concentrations of organics (org), sulphate, nitrate, ammonium and chloride of PM with vacuum aerodynamic diameters smaller than 1 μm ($PM_1$). The instrument is described in detail elsewhere (e.g. Ng et al., 2011). In brief, ambient particles are sampled through an aerodynamic focusing lens (AFL). The particles in the narrow beam impact onto a vaporizer heated at ~600℃ and volatilize. The vapors are then ionized and analyzed

by a quadrupole mass spectrometer.





For QA/QC, the ACSM was calibrated for its inlet flow rate, the ionization efficiency of nitrate (response factor, RF), and the relative ionization efficiencies (RIE) of ammonium and sulphate (for the winter campaign only) before, during and after the campaigns. Monodispersed $NH_3NO_3$ and $(NH_4)_2SO_4$ particles of 300 nm were used for the ionization calibration using an atomizer (3076, TSI), a differential mobility analyzer (DMA, 3081, TSI), and a condensation particle counter (CPC, 3786, TSI).

Additionally, an internal naphthalene source and $N_2$ beam signals were used to adjust for the system stability (i.e. secondary electron multiplier decay). Blank tests were also conducted by introducing filtered air for 5 days and were used to validate the stability and quantify the detection limit for each species. Details and discussion on the ionization efficiencies, DLs, analytical uncertainties, and the collection efficiency of the ACSM can be found in the Supplement and Table S1.

As described in the Supplement, the total species concentration of organics, sulphate, nitrate, and ammonium by the

ACSM was comparable to the $PM_{2.5}$ mass measured by a $PM_{2.5}$ monitor (SHARP 5030, Thermo Sci. Inc.). The reconstructed mass from the ACSM and the Xact data showed good correlation and agreement with the measured $PM_{2.5}$ mass concentration (Figs. S1, S2, and Table S2).

**2.4 Black carbon and supporting measurements**

One minute black carbon concentrations were measured using a 7 wavelength Aethalometer (AE33, Magee Scientific). The AE33 measures the light absorption of dual sampling spots, which are loaded at different flow rates. The measurement of the different loading and attenuation in the AE33 is used to compensate the filter loading effect resulting in linear reduction of the instrumental response as particle loading increases (Drinovec et al., 2015). In this study, hourly averaged Aethalometer BC data

using a wavelength of 880 nm were used as a refractory carbonaceous material in the source apportionment modeling.

Gaseous pollutants (nitrogen oxides ($NO_x$), carbon monoxide (CO), and sulphur dioxide ($SO_2$)) and ultrafine particles (UFP) were measured using gas analysers (42i, 48i, 43i, Thermo Scientific) and a fast mobility particle sizer (FMPS 3091, TSI), respectively. Traffic counts were measured using a traffic sensor (SS125, SmartSensor HD, Wavetronix) for the cold month campaign. The SmartSensor HD utilizes a dual-radar system to detect vehicles' lengths, number, and speeds. Mass

concentrations of $PM_{2.5}$ were measured by a synchronized hybrid ambient real-time particulate monitor (SHARP 5030, Thermo Scientific) monitor at the 4[th] floor rooftop of the SOCAAR lab. Wind direction, wind speed, ambient temperature and relative humidity were also measured using a WeatherHawk meteorological sensor (model 500, WeatherHawk) on the rooftop. Table S3 gives an overview of the monthly average concentrations and standard deviations of the collocated measurements. Wind rose plots for the two campaign periods and the diurnal variations of these supporting parameters are shown in the Supplement (Figs.

S3, S5, and S6).

**2.5 Application of PMF**

Positive matrix factorization (PMF) is one of the most common receptor models used in the identification of PM sources in the

atmosphere and their contributions. Details are described elsewhere (Paatero and Tapper, 1994; Paatero, 1997). In brief, the non-negative factor analysis model is applied to a measured data matrix $\mathbf{X}$ ($n \times m$), where $n$ is the number of samples and $m$ is the number of chemical species, to solve for two matrices $\mathbf{G}$ ($n \times p$) and $\mathbf{F}$ ($p \times m$) as well as a residual matrix $\mathbf{E}$, where $p$ is the number of sources and $\mathbf{E}$ is the unexplained portion of $\mathbf{X}$. A least-squares algorithm is used to iteratively solve for the matrices by minimizing the object function (Q) defined as follows:




$$Q = \sum_{i=1}^{n} \sum_{j=1}^{m} \left( \frac{e_{ij}}{s_{ij}} \right)^2$$

(1)

where $e_{ij}$ is an element in the $n \times m$ matrix, **E**, of residual concentrations and $s_{ij}$ is the analytical uncertainty of the $j^{th}$ species in the $i^{th}$ sample. In this study, PMF analyses were performed using EPA PMF 5.0 which incorporates the multilinear engine (ME-2) (Norris et al., 2014). The preparation of input data and uncertainty matrices for trace metals, inorganic ions, OA mass spectra, and BC are described in the Supplement.

Three PMF scenarios were investigated in this study in order to compare and contrast PM source apportionment solutions using available chemical speciation data. One scenario involved the application of PMF to Xact metal concentrations only (PMF$_{metal}$, 4234×16). The second scenario used ACSM organic mass spectra only (PMF$_{org}$, 4234×121). The last scenario used combined data set including metals, the ACSM organic mass spectra, inorganic species from the ACSM (i.e. sulphate, nitrate, and ammonium), and Aethalometer BC (PMF$_{Full}$, 4234×141). In each PMF analysis, all available data for all seasons (warm and cold months) were combined.

The number of possible sources ($p$) was initially determined by investigating the ratio of Q to "Q$_{exp}$" which is the theoretical value of Q defined by the equation, $n \times m$ - $p \times (n + m)$. Only good species described in the Supplement were taken into account when calculating the Q/Q$_{exp}$ ratio. A smaller change in Q/Q$_{exp}$ with increasing factor numbers may indicate that the increasing number of factors is not meaningful. The range of possible solutions were examined to see if one solution was more physically realistic than the others. In addition, the robustness of the factor analysis was evaluated by exploring an FPeak rotational analysis which increases or decreases elements of the matrix **G** or **F**. The solutions produced by rotations across FPeak values from -5 to 5 using the EPA PMF application were examined.

EPA PMF 5.0 includes useful tools to estimate uncertainties and evaluate the robustness and rotational ambiguity of PMF modeling results. One of the new features of the EPA PMF version 5.0 is an additional tool to evaluate the rotational uncertainty of a PMF solution (i.e. displacement analysis). With the displacement analysis, each element in source profile is displaced from it fitted value in a PMF solution to estimate the uncertainties for each element in each factor profile. Based on the result of the displacement analysis of a PMF solution, the rotational ambiguity of PMF solutions was assessed (i.e. number of swaps at the lowest predetermined Q levels). Bootstrap analysis was performed to quantify the uncertainty of a PMF-resolved solution. In addition, 100 bootstrap iterations were conducted to obtain the percentage of factors assigned to each base case factor (i.e. bootstrap mapping) and determine unstable factors in PMF solutions.

In order to determine the source contribution to PM$_{2.5}$, the reported contributions of the PMF$_{Full}$ factors were regressed against the measured PM$_{2.5}$ mass concentrations using multiple linear regression (MLR). In the PMF$_{metal}$ and PMF$_{org}$ analysis, total metal concentrations and total organic concentrations were used instead for the MLR, respectively. The temporal trends of PMF resolved factors were examined for diurnal trends, weekday/weekend differences, and seasonal trends, and compared with collocated supporting measurements (i.e. gaseous pollutants, particle number concentrations, traffic volumes and meteorological parameters). Spearman's rank-order correlation coefficients (r) between the variations of PMF source contributions and the supporting measurements were estimated to provide further useful information for the characteristics of sources. A p-value of < 0.01 was used to assess the statistically significance in the correlation analysis. In order to identify the most probable direction of local sources, the conditional probability function (CPF, Ashbaugh et al., 1985; Jeong et al., 2011) was evaluated for each source by estimating the probability that high source contributions of PMF-resolved sources are related to specific wind directions. These analyses provided further insight into the nature of the PMF-resolved factors.



## 3 Results and discussion

### 3.1 PMF of combined data, PMF$_{Full}$

### 3.1.1 Selection of PMF solution

PMF analysis was applied to the combined data including hourly organic mass spectra and inorganic species from the ACSM, metal concentrations from the Xact, and Aethalometer BC data (i.e. a total of 141 variables) collected over the entire measurement period (i.e. 4234 samples). A summary of the concentrations and diurnal variations of these data are presented in the Supplement (Table S1, Fig. S4). In the PMF$_{Full}$ analysis, the most reliable solution was explained by nine factors. The nine factor solution included five local factors (i.e. Road Dust, Primary Vehicle Emissions, Tire Wear, Cooking, and Industrial

Sector) and four regional factors (Biomass Burning, Oxidised Organics, Sulphate and Oxidised Organics, and Nitrate and Oxidised Organics). The chemical profiles and average contributions of the nine factors are shown in Fig. 1 and Table 1 with the time series of these factors included in the Supplement (Fig. S7).

Four to thirteen factor solutions were initially examined, from which possible solutions (i.e. eight to ten factor solutions) were chosen based on the change of Q/Q$_{exp}$, the achievement of the constant and global minimum of Q, the Fpeak rotational

analysis, the displacement of factor elements, and the interpretation of physically meaningful factors. In the eight-factor solution, the Road Dust and Primary Vehicle Emissions factors could not be fully resolved, whereas the ten-factor solution split the Primary Vehicle Emissions into a Cu-Ba factor and an unstable HOA factor. In moving from eight to ten factors, there was no clear decrease rate in Q/Q$_{exp}$ as shown in Fig. S8. However, the nine-factor solution showed the most stable global minimum of Q over 100 random runs. There was no factor swap in the displacement analysis with the nine-factor solution and little increase

in Q/Q$_{exp}$ in the Fpeak analysis (Q/Q$_{exp}$ of < 0.3% within -5 to 5, Fig. S8), indicating that the solution was well-defined and rotationally robust. In terms of bootstrap analysis, all factors except for the Biomass Burning factor were stable with the eight- and nine-factor solutions. However, the Biomass Burning factor was less stable with the eight-factor solution (mapped on 51% of runs) as compared to the nine-factor solution (mapped on 68% of runs). The robustness of the nine-factor solution in the PMF$_{Full}$ analysis was reinforced by the combination of the higher reproducibility of the Biomass Burning factor and little rotational

ambiguity in the solution.

In many AMS/ACSM OA studies, the degree of oxidation and volatility has been represented by the fraction with mass-to-charge ratio (m/z) of 44 ($CO_2^+$) to the total OA signals (i.e. F44), which increases with photochemical age (e.g. Aiken et al., 2008; Ng et al., 2010). Conversely, the ratio of m/z 43 ($C_3H_7^+/C_2H_3O^+$) to the total organics (i.e. F43) is indicative of less oxidised OA. In this study, F44 and F43 were considered as indicators of atmospheric aging for PMF-resolved factors. The

ranges of F44 and F43 for each factor in the PMF$_{Full}$ analysis are displayed in Table 1.

The diurnal and monthly contributions of the nine factors are depicted in Figs. 2 and 3. The correlations between the factors and collocated measurements including UFP, gases pollutants, and vehicle counts are shown in Table 2. The CPF plots for the nine factors are depicted in Fig. S9. These further analyses helped characterise the PMF-resolved factors so as to identify their probable sources and name them.


### 3.1.2 Road Dust

The Road Dust factor was characterized by mineral elements (i.e. Ca, Fe, Si, Ti) suggestive of soil (Fig. 1). The factor exhibited a strong diurnal trend and was notably higher on weekdays than weekends (Fig. 2). This weekday high pattern and a higher

correlation with vehicle counts clearly indicated that this factor was associated with local anthropogenic activities such as traffic



(Tables 1 and 2) rather than being wind entrained soil particles. However, the midday peak in the diurnal pattern was not consistent with traffic alone and the crustal elements in its composition indicated that it was a mix of abrasion particles from vehicles and soil dust characterized by Al, Si, Ca, and Fe (e.g., Viana et al., 2008; Amato et al., 2016). A strong inverse correlation of the Road Dust factor with relative humidity and a higher contribution in drier months (e.g. April and May in Fig.

3) indicated that its emissions were also promoted by favorable meteorological conditions, perhaps the drying of the pavement over the day, in agreement with the seasonal variation of a crustal matter factor (Al-Si-Ca) observed by Sofowote et al. (2015) in Toronto. The Road Dust factor was 64% organic aerosol, accounting for approximately 8 % of the total organic aerosol concentration. Amato et al. (2014) reported a high contribution of organic carbon in urban road dust samples. Interestingly, a high fraction of m/z 44 (F44=0.22) was observed for the Road Dust factor, suggesting that road dust is a source of highly aged

OA. This high degree of oxidation implied that road dust particles accumulate organic aerosol components, perhaps by being repeatedly re-suspended by passing vehicles until they are removed after several days from the road surfaces by precipitation or street cleaning. Increased resuspension over the day, as the pavement warms and the relative humidity drops, along with accumulation of OA mass, would also explain the midday peak in this factor's diurnal pattern. In summary, this factor was attributed to traffic-induced resuspension of road dust particles consisting of soil particles, particles from vehicles, and organic

aerosol from atmospheric processing. Identification of road dust as a potential source of aged organic aerosol illustrated an advantage made possible by performing the PMF$_{Full}$ analysis.

### 3.1.3 Primary Vehicle Emissions

The Primary Vehicle Emissions factor was distinguished by high loadings of m/z 57, 43, 69, 71, 85, and 97 which are typical hydrocarbon fragments; m/z 57 (C$_4$H$_9^+$) is a key marker for hydrocarbon-like organic aerosol (HOA). The majority of the Cu, Ba, and BC concentrations were apportioned to this factor. The Primary Vehicle Emissions factor contributed 8% to PM$_{2.5}$, whereas this traffic-related factor accounted for 36% and 13% of BC and m/z 57, respectively, with the highest F57 of all factors (fraction of m/z 57 to the total organic signal in the factor profile). A weekday-high pattern and high correlations with UFP, NOx, CO, and

vehicle counts supported a strong influence of traffic emissions on the Primary Vehicle Emissions factor. The BC and HOA associated with this factor were suggestive of tail-pipe emissions, however, the specific source of the Ba and Cu was less clear. Emissions of Ba and Cu can arise from both vehicular exhaust, including lubrication oil/fuel combustion (e.g. Lin et al., 2005; Ntziachristos et al., 2007), and non-tailpipe emissions caused by abrasion of brakes (e.g. Councell et al., 2004; Lough et al., 2005; Schauer et al., 2006; Johansson et al., 2009; Pérez et al., 2010; Apeagyei et al., 2011; Song and Gao, 2011; Harrison et al.,

2012). Dall'Osto et al. (2013), for example, found a comparable brake dust factor characterized by Cu and Fe with a strong diurnal pattern. Although it remains unclear as to whether the metals in this factor reflected mixing-in of non-tailpipe emission, the presence of the BC and HOA indicated at least some contribution from tail-pipe emissions. As shown in Fig. 2, the diurnal variations of the Primary Vehicle Emissions factor and the Road Dust factor were found to be different, with only the former showing a narrow dominant peak during the morning rush hour. This distinct diurnal trend of the Primary Vehicle Emissions

factor was suggestive of direct vehicle emissions (vs. resuspension) and highlighted that the separation of Primary Vehicle Emissions and Road Dust factors in the nine-factor solution was more appropriate.

### 3.1.4 Tire Wear





Similar to the Primary Vehicle Emissions factor, the Tire Wear factor was also characterized by a strong diurnal peak coinciding with the morning rush hour and higher weekday contributions. The m/z 43 is typical of fragments of $C_3H_3O^+$ and $C_3H_7^+$ from saturated hydrocarbons, aldehydes, and ketones. The highest F43 were observed for the Primary Vehicle Emissions and Tire Wear factors, consistent with these factors being traffic related (Table 1). In contrast to the Road Dust and Primary Vehicle

Emissions factors, the contribution of the Tire Wear factor was higher in the winter (January-February) than the average in the spring (April-May) by a factor of 1.5 (vs. 0.40 for Road Dust, 0.32 for Primary Vehicle Emissions). The element Zn and Mn are often associated with tire wear dust (e.g. Wahlin et al., 2006; Apeagyei et al., 2011; Pant and Harrison, 2013). It is interesting to note that despite the strong correlation between the Tire Wear factor and traffic-related pollutants (i.e. UPF, NOx, and CO), a weaker correlation was observed with vehicle counts (Table 2). It is possible that the Tire Wear factor contribution is more

influenced by driving conditions (i.e. increased tire wear during heavy accelerating and braking) and road conditions rather than just the number of vehicles. Furthermore, during the cold period the diurnal trends of the Tire Wear factor showed a minor peak at around 3 am when traffic volume is typically the lowest; snow plows in winter and garbage trucks, are the main vehicles on Toronto roads at this hour of the morning. Thus, this minor peak may have been due to this additional winter traffic. Finally, as compared to the other two traffic-related factors, a more evident south-southwest directionality was observed in the CPF plot of

the Tire Wear factor (Fig. S9c). This directionality was consistent with the location of the closest intersection and the most congested section of an expressway as well as a city airport, all located 3 km southwest of the monitoring site. In summary, most of the available evidence indicated that the Tire Wear factor was mostly traffic related. However, one contradiction was evident. The Tire Wear factor's higher correlation (r=0.37, p <0.01) with $SO_2$ was not consistent with traffic given the low sulphur fuel used in Canada. Closer investigation indicated that the measurement site was also being affected by occasional short term spikes

with high Zn and $SO_2$ suggesting that mixed into this factor were plumes from an unknown local industrial source. No plausible source could be identified and the origin of these plumes is still under investigation. In summary, the PMF$_{Full}$ analysis allowed identification of a factor that was believed to originate mostly from tire wear. Mixed in with this were plumes from another source that could be recognised with the hourly time resolution data but not separated into a distinct stable factor.

**3.1.5 Industrial Sector**

As a minor source, the Industrial Sector factor contained high loadings of Pb and As, and accounted for 2% of PM$_{2.5}$. While the As concentration was below detection most of the time, the episodes of elevated As coincided with spikes in the occurrence of this factor. No diurnal, weekday, or seasonal patterns were observed for the Industrial Sector factor. Atmospheric Pb and As can

arise from non–ferrous metals smelting and may be indicative of municipal waste incineration (e.g. Zhang et al., 2009). The high loading (20%) of BC in this factor implied the influence of fossil fuel combustion on this factor. The east-southeast directionality observed in the CPF (Fig. S9d) was consistent with the location of a once heavily industrialised sector of Toronto with soil contaminated by a range of pollutants, including Pb. This sector also includes a wastewater treatment facility, located 6 km to the east-southeast of the monitoring site. In summary, the PMF$_{Full}$ analysis was able to identify the existence of a previously

unrecognised source of Pb and As, along with a potential location.

**3.1.6 Cooking**

The Cooking factor was characterized by m/z 55 and m/z 57 with distinct peaks in its diurnal pattern at noon and 20:00. The

evening peak was much larger on weekends, consistent with increased activity of restaurants. This factor explained the second





largest fraction of organic aerosol (15%). Separation of cooking and hydrogen-like organic aerosol factors is often difficult in PMF analyses applied to organics only (i.e. resolution of COA and HOA). In this $PMF_{Full}$ analysis, inclusion of additional variables (i.e. trace metals, BC) was advantageous in that it facilitated this separation and enhanced the overall apportionment of organic aerosol factors, as further discussed in the $PMF_{org}$ analysis. The resulting F55 (the m/z 55 signals normalized by the total

organics) for the Cooking factor was higher than the F57 by a factor of 2, whereas a much lower F55/F57 ratio was observed for the Primary Vehicle Emissions factor (F55/F57=0.7) as compared to the $PMF_{org}$ results.

### 3.1.7 Biomass Burning

The Biomass Burning factor was characterized by K, m/z 60 ($C_2H_4O_2^+$) and m/z 73 ($C_3H_5O_2^+$), all known markers for wood burning (Alfarra et al., 2007). The Biomass Burning factor displayed the highest F60 (fraction of m/z 60 to the total organic signal in the factor) of all factors from the $PMF_{Full}$ analysis. Predominant peaks at m/z 57 and high m/z species with a lower F44 indicated that the factor was less oxygenated. A F44 of 0.06 for the Biomass Burning factor is consistent with the finding of Adler et al. (2011) and Bougiatioti et al. (2014), who reported that m/z 44 contributed approximately 6% and 4% of the total

organic aerosol during biomass burning events, respectively. From June 14 to July 4, 2014, air quality in Toronto was affected by strong biomass burning emissions due to wildfires in Quebec (Healy et al., 2015). The time series of the Biomass Burning factor exhibited the strongest impact from July 1 to 3, 2014 with a maximum concentration of approximately 20 μg/m³ (Fig. 5b). In the winter months, the Biomass Burning factor had higher contributions at night on weekends, indicating the possible influence of residential wood burning in wintertime.

### 3.1.8 Oxidised Organics

The Oxidised Organics factor was characterized by dominant signals for fragments at m/z 18, 29, and 44, all indicators of oxidised organics. The Oxidised Organics factor was composed almost entirely of organic aerosol (97%) and accounted for 18%

and 35% of $PM_{2.5}$ and the total organic aerosol mass, respectively (Table 1, Fig. 4). Compared with mass spectral profiles from other studies, an F44 of 0.20 for the Oxidised Organics factor in this study was higher than the average F44 (0.07) for SV-OOA and more comparable to the average ratio (0.17) for LV-OOA reported for multiple sites by Ng et al. (2010). The high F44 for the Oxidised Organics factor was comparable to the F44 of the Sulphate and Oxidised Organics and Nitrate and Oxidised Organics factors from the $PMF_{Full}$ analysis, whereas the F44/F43 ratio (2.3) was lower for the Oxidised Organics factor than the

ratios for the Sulphate and Oxidised Organics (3.2) and Nitrate and Oxidised Organics (8.2) factors. The presence of m/z 43 in the Oxidised Organics factor implied that it still contained primary organic aerosol. The time series of the Oxidised Organics factor from the $PMF_{Full}$ analysis was dominated by the elevated concentrations during the Quebec biomass burning event from June 14 to July 4 (Figs. S7 and 5a). During this forest fire event, the Oxidised Organics and Biomass Burning factors were the largest contributors to $PM_{2.5}$, accounting for 53% and 33% of the $PM_{2.5}$ mass, respectively, whereas their contributions during

non-event days in the warm months were 21% and 7%, respectively. In the cold months (Dec-Feb), the Oxidised Organics factor accounted for only 6% of the $PM_{2.5}$ mass. Photochemical production of oxidised organic aerosol during both biomass burning and non-event days in the warm months was likely the dominant process governing the contribution and profile for the Oxidised Organics factor, so this factor was named accordingly.

Even though the Biomass Burning and Oxidised Organics factors occurred simultaneously during the biomass burning

episodes, they did exhibit some key differences in the factor profiles (Fig. 5). In addition to the presence of K and BC in the





Biomass Burning profile, the organic fragment ratios exhibited clear differences. In terms of the oxidation degree between two factors, the F44 of the Oxidised Organics factor was higher than that for the Biomass Burning factor (0.20 vs. 0.06), indicating that the aerosol in the Oxidised Organics factor was more oxidised. The Oxidised Organics factor also showed a stronger correlation with ambient temperature relative to the Biomass Burning factor (r=0.65 vs. r=0.24). These results suggest that

during the forest fire event, the Oxidised Organics factor represented aerosol mass from the Biomass Burning factor that had been processed through photochemical aging. Adler et al. (2011) reported an OOA factor with a high F44 during a day following a massive forest fire event due to further oxidation processes. Furthermore, the organic signal profile of the Oxidised Organics factor in this study was highly comparable to the mass spectrum profile of the OOA-BB factor previously observed during a massive forest fire event in summer (Bougiatioti et al., 2014). The OOA-BB factor reported by Bougiatioti et al. (2014) was

described as an intermediately oxidised BBOA factor and exhibited a similar temporal variability with a BBOA factor.

### 3.1.9 Sulphate and Oxidised Organics

The Sulphate and Oxidised Organics factor was characterized by sulphate, ammonium, selenium, and m/z 44, contributing 17%

of the $PM_{2.5}$ mass. The element of Se is a tracer for emissions from coal combustion. This factor contained 11% of the total measured organic aerosol and 14% of BC during the study period. A high correlation with $SO_2$ and no diurnal pattern were found, implying that this factor was influenced by emissions from regional scale coal combustion. Higher concentrations of the Sulphate and Oxidised Organics factor were also observed during the cold months. The F44 of 0.21 and the ratio of F44/F43 (3.2) were comparable to values for LV-OOA found across the Northern Hemisphere (Ng et al., 2010). Previous source

apportionment studies found that in 2007 a secondary sulphate factor contributed 35% of the $PM_{2.5}$ in this urban site (Jeong et al., 2011), a much higher contribution than the 17% in 2013-14 reported here. Toronto was previously affected by emissions from local coal-fired power plants and more distant plants located to the southwest in Ohio, Indiana, and Illinois, USA (Jeong et al., 2011; Jeong et al., 2013). McGuire et al. (2014) also reported similar geographical origins of Sulphate and Oxidised Organics in wintertime using full mass spectra of both inorganic and organic components in a PMF analysis in Windsor, Canada. Results

from this study suggest that the impact of coal on $PM_{2.5}$ in Toronto has decreased substantially. Consistent with the phase-out of coal based electricity generation in Ontario, it is more likely that most of the remaining sulphate now originates from upwind states. The CPF plot of the Sulphate and Oxidised Organics factor highlights a strong association with emissions from the south-southwest (Fig. S9h). Furthermore, the high degree of oxidation observed in the Sulphate and Oxidised Organics factor was consistent with the influence of regional sources and aging processes during long-range transport.

### 3.1.10 Nitrate and Oxidised Organics

The Nitrate and Oxidised Organics factor was mainly composed of nitrate, ammonium, and m/z 44.  It was the largest contributor to $PM_{2.5}$ mass, accounting for 30%, on average. The highest F44 (0.26) was found for the Nitrate and Oxidised

Organics factor, while it also had the lowest F43 among the major oxygenated organic aerosol sources (i.e. Oxidised Organics, Sulphate and Oxidised Organics, Nitrate and Oxidised Organics). The ratio of F44/F43 was 8.2, much higher than the Sulphate and Oxidised Organics factor, indicating that the Nitrate and Oxidised Organics was associated with the most highly oxygenated organics. The temporal trends of the Nitrate and Oxidised Organics factor displayed multiday episodes, higher concentrations at nighttime, and in cold months, with an inverse relationship with ambient temperature (Figs. 2 and 3). The nighttime-

high/daytime-low diurnal pattern may have been due to sublimation of nitrate during the day or nighttime formation with



condensation. The average contribution of the Nitrate and Oxidised Organics factor sharply increased to 57% during the winter months. The presence of highly oxidised organics, highest correlation with $SO_2$ (r=0.61), and a strong CPF directionality implied that the Nitrate and Oxidised Organics factor was more due to regional industrial sources than local emissions in winter. This finding was surprising as traffic is one of the largest sources of NOx emissions in the Greater Toronto Area, yet this factor was

higher on weekends when NOx emissions are lower (23 ppb on weekdays vs. 17 ppb on weekends). It is possible that local NOx emissions did contribute some nitrate to the Nitrate and Oxidised Organics factor mass, by nighttime oxidation and condensation of locally emitted NOx, but that the oxidising atmosphere that promotes conversion of NOx to nitrate was associated with multiday meteorological conditions brought in with regional air masses from the southwest.

**3.2 Comparison of results for the PMF$_{metal}$ and PMF$_{org}$ analysis**

**3.3.1 PMF of Xact metals, PMF$_{metal}$**

In the PMF$_{metal}$ analysis, a solution with eight factors was chosen with four traffic-related factors (Ca-Si, Cu-Ba, Zn-rich, Mn-rich), a biomass burning factor (K-rich), an industry factor (Pb-rich), an oil combustion factor (Ni-V), and a coal combustion

factor (Se-rich). These factors were named based on their dominant chemical components. Among solutions using three to ten factors, an 8-factor solution was the most stable with a sharper decrease in the $Q/Q_{exp}$ trend and a constant global minimum Q value among 100 random runs. The seven-factor solution was unstable and did not extract the Ni-V factor that apportioned to the Ca-Si and K factors, while the nine-factor solution split the Ni-V factor into a similar Ni-V factor and a separate Br-rich factor that did not resemble any known sources. In a six-factor solution, the Zn-rich factor mixed with the Mn-rich factor. In terms of

the stability of the PMF$_{metal}$ analysis, all factors of the 8-factor solution were reproduced in 100% of bootstrap runs, whereas K/N-Ni, Se-rich, and Pb-rich were mapped with bootstrap in 96~98% of runs for the 6-factor solution, demonstrating that the 8-factor solution was more stable.

The chemical profiles and concentrations of the 8-factor solution from the PMF$_{metal}$ analysis are included in the Supplement (Figs. S10 and S11). The average contributions of these factors from the PMF$_{metal}$ analysis are shown in Table 3.

Regression of the contributions to allow scaling to the total PM$_{2.5}$ mass was not possible so the contributions had to be expressed in terms of the total metals instead. Incorporation of the metals data into the PMF$_{Full}$ analysis did allow estimation of the contribution of metal-rich factors to PM$_{2.5}$, a further advantage of the PMF$_{Full}$ approach. The four traffic-related factors (Ca-Si, Cu-Ba, Mn-rich, and Zn-rich) demonstrated weekday-high diurnal patterns with morning rush hour peaks (Fig. S12). These traffic-related factors were also correlated with NOx, CO and UFP (Table S4), further supporting their association with traffic.

The traffic-related factors explained approximately 70% of the total metal concentration.

The factor profiles and contributions from the PMF$_{metal}$ and PMF$_{Full}$ analyses were compared (Fig. 6 and Table 5). The Ca-Si and Cu-Ba factors from the PMF$_{metal}$ analysis were comparable to the Road Dust and Primary Vehicle Emissions factors from the PMF$_{Full}$ analysis, respectively. The temporal variations of the Zn-rich and Mn-rich factors in the PMF$_{metal}$ analysis were highly correlated with the contribution of the Tire Wear factor from the PMF$_{Full}$ analysis (Table 5). From the PMF$_{metal}$ analysis,

the Mn-rich factor typically combined with the Zn-rich and Ca-Si factors in an unstable 6-factor solution, whereas the determination of independent sources for Zn-rich and Mn-rich in the 8-factor solution became problematic. From the result of the PMF$_{Full}$ analysis, elements of Zn and Mn were assigned in the Tire Wear factor with a higher degree of stability in the solution, a further advantage of the PMF$_{Full}$ analysis.

The K-rich and Pb-rich factors from the PMF$_{metal}$ analysis were comparable to the Biomass Burning and Industrial

Sector factors identified in the PMF$_{Full}$ analysis, respectively. Interestingly, a relatively lower correlation was observed for the K-





rich and the Biomass Burning factor (r=0.65), whereas the Industrial Sector and Pb-rich contributions from the two PMF analyses were highly correlated with each other (r=0.97). As discussed in the Data preparation section of the Supplement, a major fireworks event that could be an additional source of K that was excluded from the analysis. However, there were more sporadic spikes of K in summer months possibly due to local fireworks around the downtown area. In the $PMF_{metal}$ analysis, the

additional events were captured by the K-rich factor, whereas the Biomass Burning factor, characterized by both K and organics tracers (i.e. m/z 60 and m/z 73), was more likely to represent biomass burning emissions. These results highlight a further advantage of using the $PMF_{Full}$ analysis to resolve more robust factors and to more accurately quantify contributions.

Vanadium and nickel are known to be emitted from oil combustion (e.g. Chow et al., 2004; Jeong et al., 2011; Becagli et al., 2012). The temporal variations of the Ni-V factor in the $PMF_{metal}$ analysis exhibited no strong weekday/weekend

differences. However, the average concentration in January and February was approximately 50% higher than the average in June and July, indicating a probable influence of emissions from residual oil heating in winter. A relatively high correlation with $SO_2$ (Spearman r=0.4, p<0.01) in the $PMF_{metal}$ analysis further reinforces that the Ni-V factor is linked to fossil-fuel combustion emissions (Table S4). The correlations of source contributions in Table 5 indicate that the Ni-V factor and the Se-rich factor in the $PMF_{metal}$ analysis were associated with the Sulphate and Oxidised Organics factor in the $PMF_{Full}$ analysis. A comparison of

just the metals within the source profiles of the factors showed that the Ni-V factor and the Se-rich factor were highly correlated with the Sulphate and Oxidised Organics factor with Spearman coefficients of 0.88 and 0.80, respectively. A Ni-V factor could be identified in an 11-factor solution from the $PMF_{Full}$ analysis with a very poor stability in the solution. In the nine factor $PMF_{Full}$ solution, some of the Ni and V ended up mixed into the Road Dust factor, indicating that the $PMF_{Full}$ analysis did not always enable better resolution of factors. Regardless, these results indicate that the Sulphate and Oxidised Organics factor was

influenced by emissions from coal and oil combustion, particularly during the cold months.

### 3.3.2 PMF of ACSM organics, $PMF_{org}$

The PMF receptor modeling was applied to the hourly resolved organic aerosol mass spectra obtained by the ACSM. The

stability of Q over the 100 runs, the change of $Q/Q_{exp}$, scaled residuals, Fpeak runs, and G-space plots were examined to determine a reasonable solution. The most reliable solution was explained by five factors in the $PMF_{org}$ analysis. These factors were named based on dominant components and the terminology (i.e. HOA, COA, BBOA) adopted in previous receptor modeling analyses of aerosol mass spectra (e.g. Ng et al., 2010). Three primary factors, HOA, COA, and BBOA, and two secondary organic factors, OOA and LV-OOA, were identified from the $PMF_{org}$ analysis. The mass spectra and contributions of

the five-factor solution are shown in Figs. S13 and S14 and Table 4. The diurnal variations of the PMF-resolve OA factors are shown in Fig. S15. The HOA profile was characterised by higher contributions at m/z 57 and a morning rush hour peak at 08:00 in its weekday diurnal pattern variation. The contribution of HOA was typically higher on workdays than on weekends (i.e. WE/WD =0.85) in the warm months with higher correlations with UFP and CO (Table S5). However, the weekday diurnal pattern of the HOA factor also had strong peaks at noon and 19:00 (Fig. S15); other traffic related pollutants did not exhibit

peaks at these times. The mass spectra profile of the COA factor was similar to that of the HOA factor, but with higher contributions at m/z 55. The contributions of the COA factor peaked at noon and in the evening at 19:00 with higher contributions on weekends, indicating the influence from cooking emissions at lunch and dinner time from nearby restaurants in the downtown area. The presence of these same peaks in the diurnal trend of the HOA factor suggests that the $PMF_{org}$ analysis did not properly separate the HOA and COA factors.





The PMF$_{org}$ and PMF$_{Full}$ factors were compared and exhibited consistent characteristics (Table 5 and Fig. 7). Good agreement was found in the mass spectra profiles between the PMF$_{Full}$ Primary Vehicle Emissions factor and the PMF$_{org}$ HOA factor, with a Spearman correlation coefficient (r) of 0.83. However, the intensities of m/z 57 and m/z 43 signals of the Primary Vehicle Emissions factor were enhanced in the PMF$_{Full}$ vs. the PMF$_{org}$ analyses, implying a better apportionment of hydrocarbon-

like fragments. In addition, the diurnal variation for the Primary Vehicle Emissions factor was consistent with the dominant morning traffic pattern without the noon and evening peaks (Fig. 2). These results emphasize a further advantage of including the metal speciation data in the PMF$_{Full}$ analysis: the COA and HOA factors were more effectively separated.

The PMF$_{org}$ BBOA factor was characterized by strong m/z 60 and m/z 73 intensities, marker fragments of wood burning. As with the PMF$_{Full}$ solution, the forest fire event was the dominant source that enabled resolution of the BBOA and

OOA factors in the PMF$_{org}$ analysis, which were comparable to the Biomass Burning and Oxidised Organics factors, respectively, from the PMF$_{Full}$ analysis. In addition to good correlations in the time series of the OOA and Oxidised Organics factors between the PMF$_{org}$ and the PMF$_{Full}$ analyses in Table 5, the mass spectra of these factors from the two PMF analyses compared well (r=0.81). However, there was also a key difference between the analyses: the PMF$_{Full}$ analysis achieved greater separation of highly oxygenated OA fragments than the PMF$_{org}$ analysis. Specifically, the PMF$_{Full}$ Oxidised Organics factor

contained more enhanced signals at m/z 18 and 44 than the PMF$_{org}$ OOA factor (Fig. 7). In contrast, the signals of m/z 18 and 44 in the PMF$_{Full}$ Biomass Burning factor mass spectra were lower than those for the PMF$_{org}$ BBOA factor. Since the OOA factor otherwise had characteristics of more oxidised OA as compared to BBOA, this is where the enhanced intensities of these highly oxygenated OA fragments would be expected. These results illustrate another advantage of the PMF$_{Full}$ analysis: including the trace metals (i.e. K) in the PMF$_{Full}$ analysis provided additional temporal variability to more fully separate biomass burning vs.

photochemical processing related organics.

The LV-OOA factor, the largest organic source, comprised 38% of the total organic aerosol in the PMF$_{org}$ analysis. It contained the highest fraction of oxygenated species; 62% of m/z 44 was apportioned to this factor. No distinct diurnal pattern was observed for LV-OOA, suggesting the influence of regional scale sources or processes. It was not possible to identify specific sub-categories for LV-OOA with only the organic data. As displayed in Fig. 7, the mass spectral profile of LV-OOA

from the PMF$_{org}$ analysis was highly correlated with those of three factors: Nitrate and Oxidised Organics (r=0.80), Sulphate and Oxidised Organics (r=0.78), and Road Dust (r=0.75) factors from the PMF$_{Full}$ analysis. During two campaign periods, the time series of the PMF$_{org}$ LV-OOA showed moderate correlations with the PMF$_{Full}$ Sulphate and Oxidised Organics (r=0.56) and Nitrate and Oxidised Organics (r=0.54) factors, whereas correlation between the LV-OOA factor and the Road Dust factor was weak (Table 5). Fig. 8b presents an excellent correlation between the cold month contributions of PMF$_{org}$ LV-OOA and PMF$_{Full}$

Nitrate and Oxidised Organics (r=0.90) factors. However, a considerably lower correlation was found between these two factors in the warm months (Fig. 8a). Conversely, a slightly higher correlation was observed between the Sulphate and Oxidised Organics factor time series from the PMF$_{Full}$ analysis and the PMF$_{org}$ LV-OOA factor (r=0.61) in the warm compared to cold months. These results indicate that the sources of LV-OOA could be effectively resolved by the inclusion of inorganic ions and metal species. Moreover, a split of correlations in Fig. 8c was due to the elevated LV-OOA concentrations during the Quebec

forest fire event. This suggests that the PMF$_{org}$ analysis did not completely resolve the LV-OOA factor from the BBOA and OOA factors. These results highlight a final advantage: the PMF$_{Full}$ analysis allowed for further apportionment of LV-OOA through the incorporation of inorganics and trace metal data in the analysis.

## 4    Summary and Conclusions






Hourly resolved PM speciation data including organic aerosol mass spectra, inorganic species, trace metals, and BC obtained by high time resolution instruments were used for the source apportionment analysis of $PM_{2.5}$ in a downtown Toronto, Canada during two campaign periods. In order to elucidate the effectiveness of the $PMF_{Full}$ analysis, separate PMF analyses using only the trace metals and only organic mass spectra were performed and compared. From the $PMF_{Full}$ analysis, nine factors were

identified: five local factors (i.e. Road Dust, Primary Vehicle Emissions, Tire Wear, Cooking, and Industrial Sector) and four regional factors (i.e. Biomass Burning, Oxidised Organics, Sulphate and Oxidised Organics, and Nitrate and Oxidised Organics). The traffic-related sources, accounting for 17% ($1.5\pm1.4$ µg/m³) of $PM_{2.5}$, showed strong peaks during the morning rush hour with clear weekend/weekday differences. Strong seasonal patterns were observed for the traffic-related emissions; higher contributions of Road Dust in spring vs. a winter high pattern for Tire Wear. The three types of traffic-related emissions

described 14% of the total organic aerosol mass, whereas 83% of total metals and 49% of BC were explained by traffic. Nitrate coupled with highly oxidised organics was the largest contributor to $PM_{2.5}$ in the urban area, accounting for 30% of $PM_{2.5}$ on average, with a strong nighttime diurnal pattern and the highest monthly concentration in February. Biomass Burning contributed the majority of the $PM_{2.5}$ mass (52%) in June and July due to a major forest fire event. Much of this mass was due to photochemical aging of the biomass burning aerosol. The $PMF_{Full}$ analysis allowed a detailed apportionment the organic mass.

The highest portion was in a Oxidised Organics factor (35%), followed by Cooking (15%), Biomass Burning (12%), Sulphate and Oxidised Organics (11%), and Nitrate and Oxidised Organics (11%). The Nitrate and Oxidised Organics (F44=0.26) and Sulphate and Oxidised Organics (F44=0.21) factors contained the most highly oxidised organic aerosol, indicative of regional contributions, especially in winter.

Comparisons of the $PMF_{Full}$ analysis with $PMF_{metal}$ and $PMF_{org}$ showed many advantages of combining the high time

resolution data in terms of enabling i) the quantification of metal-rich sources, ii) the resolution of more robust factor profiles and contributions, and iii) the identification of more sources of organic aerosol. In the $PMF_{Full}$ analysis, the $PM_{2.5}$ contributions of the three traffic-related factors, mostly characterised by minor trace metals, were quantified by apportioning organic aerosol, inorganics, and BC. Relatively less stable factors from the $PMF_{metal}$ analysis were attributed to more robust factors through the $PMF_{Full}$ analysis. The $PMF_{org}$ analysis could not fully resolved the HOA factor from the COA factor due to the similarity in their

contributions and mass spectra. Inclusion of additional markers (i.e. metals and BC) allowed for the clear separation of organic aerosol mass spectra and the quantification of more accurate source contributions. In addition, the temporal variabilities of inorganic ions and trace metals in the $PMF_{Full}$ analysis provided additional structure to subdivide the LV-OOA into three sources; surprisingly, road dust was identified as a potential source of LV-OOA.

In summary, the application of PMF receptor modeling to hourly resolved trace metals with organic mass spectra,

inorganic species, and black carbon in spring, summer, and winter offered many advantages in the apportionment of both metal-rich and organic-rich, local and regional sources. Furthermore, this combined analysis allowed for better apportionment of organic aerosol sources and yielded more accurate source contributions of metal-rich sources in comparison to the PMF of organic only or metal data only.





**Acknowledgements**

This work was made possible through instruments provided by the Canada Foundation for Innovation, the Ontario Innovation Trust and the Ontario Research Fund. Some operational support was provided by Environment Canada.

**Author Contributions**

C.-H. J. performed measurements, analyzed data and wrote the paper. J. M.W. was involved in the Xact trace metals and gas

10 measurements. G. J. E. was involved in the planning the measurement, interpretations of the data, and oversaw the manuscript preparation.





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



**Table 1. Source contributions (mean ± standard deviation) of factors resolved from PMFFull analysis and the fractions of organic fragments normalized by the total organics for each factor during the entire measurement period (April 5 – July 14, 2013, November 25, 2013 – February 11, 2014).**

| PMF$_{Full}$ Factor | Source Category | PM$_{2.5}$ Contribution | | WE/WD[a] | F$_{OA}$[b] | F44 | F43 | F55 | F57 | F60 |
|---|---|---|---|---|---|---|---|---|---|---|
| | | μg/m³ | (%) | | | | | | | |
| Road Dust | Traffic | 0.6±0.6 | 6 | 0.69 | 0.64 | 0.22 | 0.00 | 0.03 | 0.00 | 0.00 |
| Primary Vehicle Emissions | Traffic | 0.7±0.8 | 8 | 0.60 | 0.27 | 0.00 | 0.13 | 0.05 | 0.07 | 0.00 |
| Tire Wear | Traffic | 0.3±0.3 | 3 | 0.63 | 0.35 | 0.06 | 0.13 | 0.03 | 0.05 | 0.00 |
| Industrial Sector | Industrial | 0.2±0.2 | 2 | 1.10 | 0.36 | 0.03 | 0.01 | 0.03 | 0.02 | 0.01 |
| Cooking | Cooking | 0.8±0.9 | 8 | 1.25 | 0.88 | 0.01 | 0.04 | 0.08 | 0.04 | 0.00 |
| Biomass Burning | Biomass Burning | 0.7±1.6 | 8 | 1.05 | 0.79 | 0.06 | 0.03 | 0.05 | 0.04 | 0.02 |
| Oxidised Organics | Photochemistry | 1.7±3.0 | 18 | 1.23 | 0.97 | 0.20 | 0.09 | 0.01 | 0.00 | 0.00 |
| Sulphate Oxidised Organics | Industrial | 1.6±1.7 | 17 | 1.21 | 0.32 | 0.21 | 0.07 | 0.01 | 0.00 | 0.00 |
| Nitrate Oxidised Organics | Industrial | 2.9±4.8 | 30 | 1.30 | 0.17 | 0.26 | 0.03 | 0.01 | 0.00 | 0.00 |

[a]WE: weekends and holidays, WD: workdays, [b]F$_{OA}$: the faction of total organic aerosol apportioned to the factor



**Table 2. Correlations (Spearman r) of factors from the PMF_Full analysis with UFP, NOx, CO, SO₂, ambient temperature (Temp), relative humidity (RH), wind speed (WS), and vehicle counts (Traffic). The highest correlation coefficients for each case are denoted in bold.**

| PMF$_{Full}$ Factor | UFP | NO$_X$ | CO | SO$_2$ | Temp | RH | WS | Traffic[a] |
|---|---|---|---|---|---|---|---|---|
| Road Dust | 0.20 | -0.02 | 0.10 | 0.04 | **0.25** | **-0.40** | -0.09 | **0.25** |
| Primary Vehicle Emissions | 0.28 | 0.30 | **0.52** | -0.33 | 0.35 | -0.16 | **-0.43** | **0.40** |
| Tire Wear | **0.32** | **0.43** | 0.17 | 0.37 | -0.11 | 0.01 | 0.01 | 0.10 |
| Industrial Sector | 0.16 | **0.19** | 0.17 | **0.19** | 0.08 | 0.14 | -0.14 | 0.03 |
| Cooking | **0.33** | 0.17 | **0.54** | -0.07 | 0.33 | 0.10 | -0.18 | 0.39 |
| Biomass Burning | 0.14 | 0.08 | **0.48** | 0.12 | 0.24 | 0.25 | -0.20 | -0.21 |
| Oxidised Organics | 0.06 | -0.22 | **0.61** | -0.34 | **0.65** | 0.12 | **-0.44** | 0.04 |
| Sulphate Oxidised Organics | -0.02 | 0.02 | 0.09 | **0.37** | -0.01 | 0.36 | 0.14 | -0.11 |
| Nitrate Oxidised Organics | 0.08 | 0.41 | -0.03 | **0.61** | **-0.44** | **0.41** | 0.15 | -0.13 |

[a] Traffic count data were available for the cold months (November 2013- February 2014).





**Table 3. Source concentrations (mean ± standard deviation) of factors resolved PMF$_{metal}$ analysis during the measurement period (April 5-July 14, 2013, November 25, 2013-February 11, 2014). Total metal concentrations were 0.45 ± 0.37 µg/m³ and 0.26 ± 0.16 µg/m³ during the warm and cold measurement periods, respectively.**

| PMF$_{metal}$ Factor | Source Category | Total metal Contribution | | WE/WD[a] |
|---|---|---|---|---|
| | | µg/m³ | (%) | |
| Ca-Si | Traffic | 0.16±0.19 | 42 | 0.57 |
| Cu-Ba | Traffic | 0.06±0.09 | 18 | 0.88 |
| Zn-rich | Traffic | 0.01±0.02 | 3 | 0.62 |
| Mn-rich | Traffic | 0.04±0.04 | 10 | 0.55 |
| V-Ni | Oil combustion | 0.04±0.06 | 12 | 1.24 |
| Se-rich | Coal combustion | 0.01±0.01 | 2 | 1.10 |
| Pb-rich | Industrial | 0.05±0.03 | 12 | 1.05 |
| K-rich | Biomass Burning | 0.00±0.00 | 1 | 1.25 |

[a]WE: weekends and holidays, WD: workdays





**Table 4. Source concentrations (mean ± standard deviation) of factors from PMF$_{org}$ analysis during the measurement period (April 5-July 14, 2013, November 25, 2013-February 11, 2014). Average organics concentrations were found to be 5.5 ± 5.6 µg/m³ and 3.6 ± 2.4 µg/m³ during the warm and cold measurement periods, respectively.**

| PMF$_{org}$ Factor | Source Category | Total OA Contribution | | WE/WD[a] |
| --- | --- | --- | --- | --- |
| | | µg/m³ | (%) | |
| HOA | Traffic | 0.89±0.76 | 19 | 0.85 |
| COA | Cooking | 0.56±0.62 | 12 | 1.33 |
| BBOA | Biomass Burning | 0.37±0.93 | 8 | 1.18 |
| OOA | Photochemistry | 1.04±1.96 | 22 | 1.14 |
| LV-OOA | Industrial | 1.78±1.44 | 38 | 1.17 |

[a]WE: weekends and holidays, WD: workdays





**Table 5.** Correlation (Spearman r) of source contributions from the PMF$_{Full}$ analysis with PMF$_{metal}$ and PMF$_{org}$ analyses. The highest correlation coefficients for each case are denoted in bold.

| PMF$_{Full}$ | | Road Dust | Primary Vehicle Emissions | Tire Wear | Industrial Sector | Cooking | Biomass Burning | Oxidised Organics | Sulphate Oxidised Organics | Nitrate Oxidised Organics |
|---|---|---|---|---|---|---|---|---|---|---|
| | Ca-Si | **0.80** | 0.59 | 0.32 | 0.01 | 0.00 | -0.01 | 0.03 | -0.21 | -0.27 |
| | Cu-Ba | 0.24 | **0.90** | 0.18 | 0.03 | 0.25 | -0.03 | 0.23 | -0.29 | -0.34 |
| | Zn-rich | 0.17 | -0.04 | **0.85** | 0.53 | 0.16 | 0.46 | 0.13 | 0.39 | 0.53 |
| PMF$_{metal}$ | Mn-rich | 0.59 | 0.45 | **0.78** | 0.24 | 0.15 | 0.21 | 0.09 | 0.10 | 0.16 |
| | K-rich | 0.21 | 0.18 | 0.31 | 0.40 | 0.38 | **0.65** | 0.52 | 0.35 | 0.23 |
| | Pb-rich | 0.14 | -0.04 | 0.37 | **0.97** | 0.25 | 0.50 | 0.35 | 0.48 | 0.42 |
| | Ni-V | 0.33 | -0.17 | 0.36 | 0.42 | 0.06 | 0.54 | 0.12 | **0.54** | 0.42 |
| | Se-rich | -0.07 | -0.14 | 0.27 | 0.38 | 0.21 | 0.39 | 0.21 | **0.66** | 0.38 |
| | HOA | 0.29 | 0.39 | 0.25 | 0.21 | **0.60** | 0.56 | 0.43 | 0.09 | 0.07 |
| | COA | -0.20 | -0.04 | 0.13 | 0.16 | **0.85** | 0.40 | 0.56 | 0.33 | 0.30 |
| PMF$_{org}$ | BBOA | -0.02 | -0.10 | 0.23 | 0.36 | 0.43 | **0.84** | 0.43 | 0.51 | 0.53 |
| | OOA | -0.05 | 0.27 | 0.09 | 0.20 | 0.55 | 0.55 | **0.88** | 0.22 | -0.01 |
| | LV-OOA | 0.12 | -0.10 | 0.20 | 0.32 | 0.46 | **0.71** | 0.68 | 0.56 | **0.54** |





**Figure 1. Factor profiles of the nine-factor solution (Road Dust, Primary Vehicle Emissions, Tire Wear, Industrial Sector, Cooking, Biomass Burning, Oxidised Organics, Sulphate and Oxidised Organics, Nitrate and Oxidised Organics) from PMF$_{Full}$ analysis including ACSM organic mass spectra, ACSM inorganics, Xact metals, and Aethalometer black carbon. Black bars on the left y-axis represent the concentration of each species apportioned to the factor. Error bars represent uncertainties estimated by 100 bootstrap runs. Red dots on the right y-axis represent the percentage of each species apportioned to the factor.**





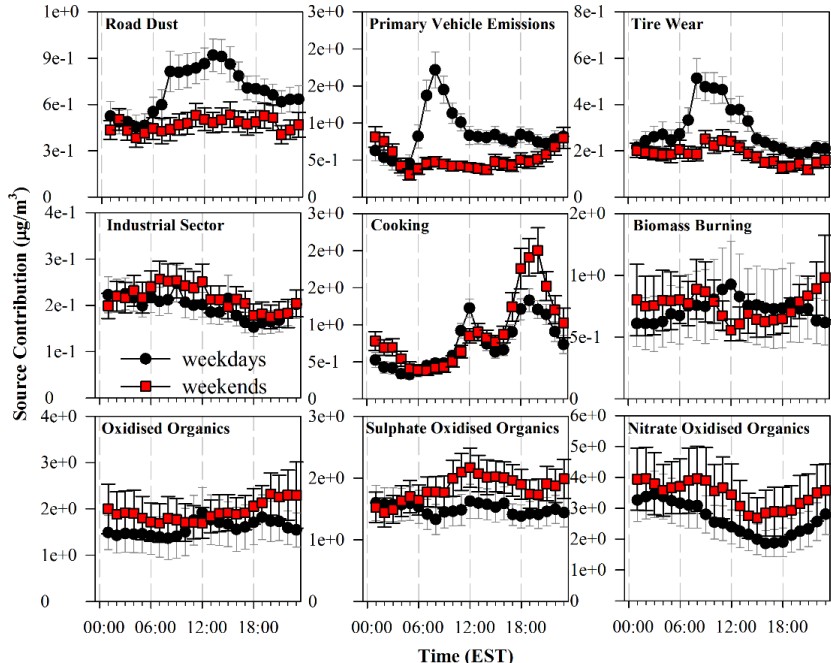

**Figure 2. Diurnal trends on weekdays and weekends during the entire measurement period (April 5-July 14, 2013, November 25, 2013-February 11, 2014) for Road Dust, Primary Vehicle Emissions, Tire Wear, Industrial Sector, Cooking, Biomass Burning, Oxidised Organics, Sulphate and Oxidised Organics, Nitrate and Oxidised Organics from the PMF_{Full} analysis. Error bars represent the 95% confidence intervals.**





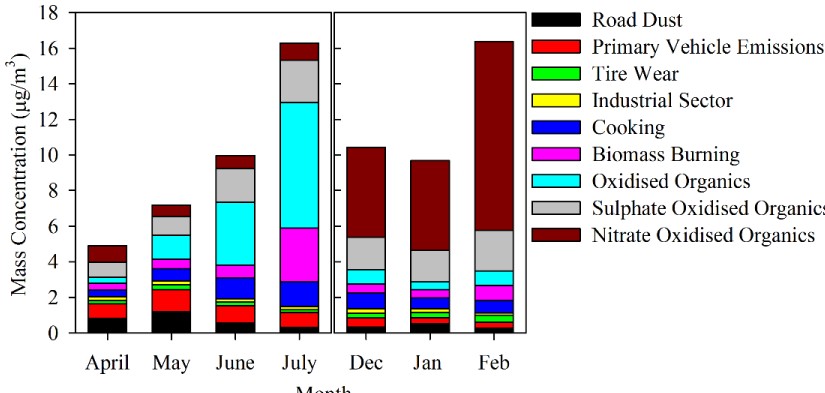

**Figure 3. Monthly variations in the contributions of the nine PMF factors from the PMF_Full analysis.**





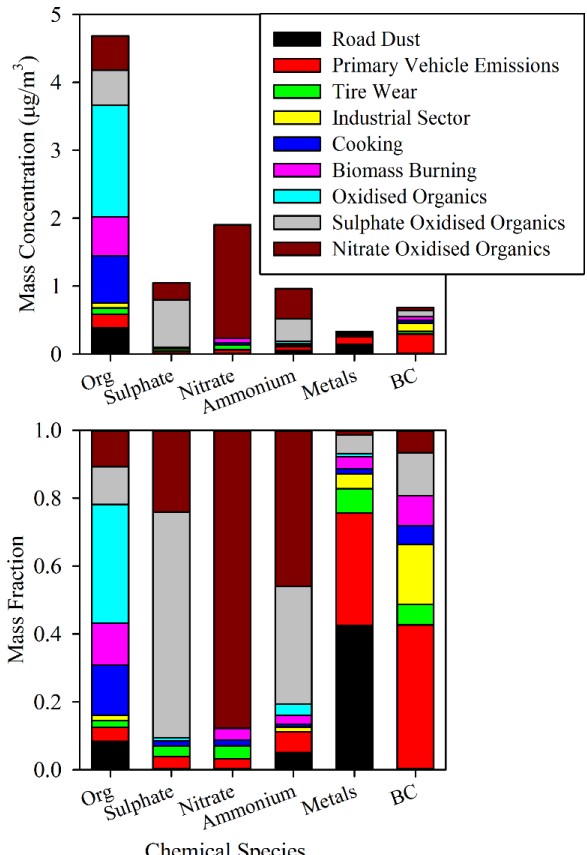

**Figure 4. Contribution of the nine factors from the PMF_Full analysis to species components, organics (org), sulphate, nitrate, ammonium, total metals, and black carbon (BC).**





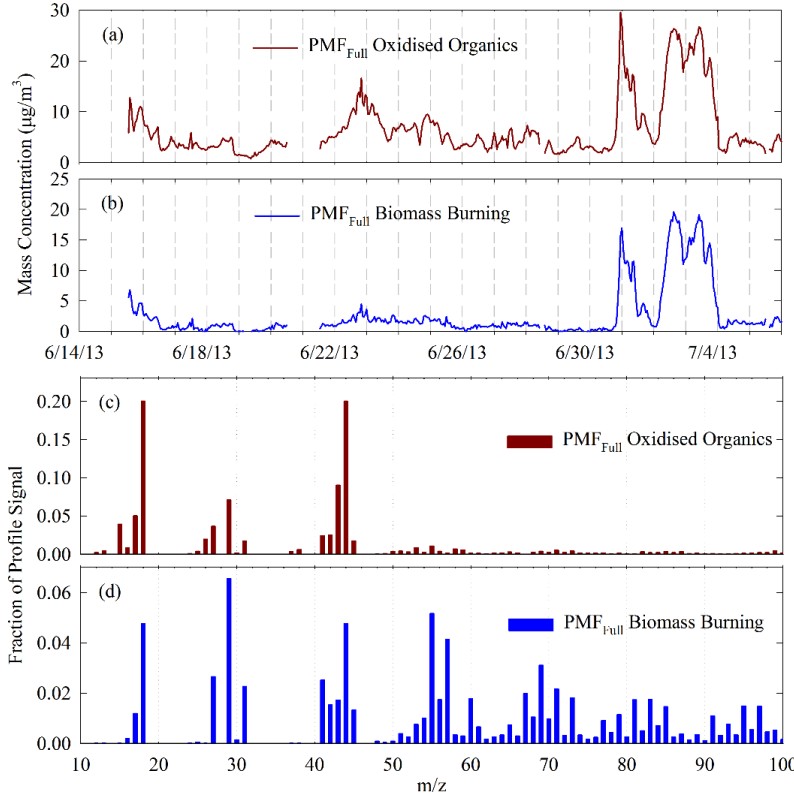

**Figure 5. Comparison of the Oxidised Organics (a, c) and Biomass Burning (b, d) factors from the PMF$_{Full}$ analysis for m/z 12-100. The bars represent the amount of each mass spectra signal apportioned to the total organic fraction signal of the factor.**



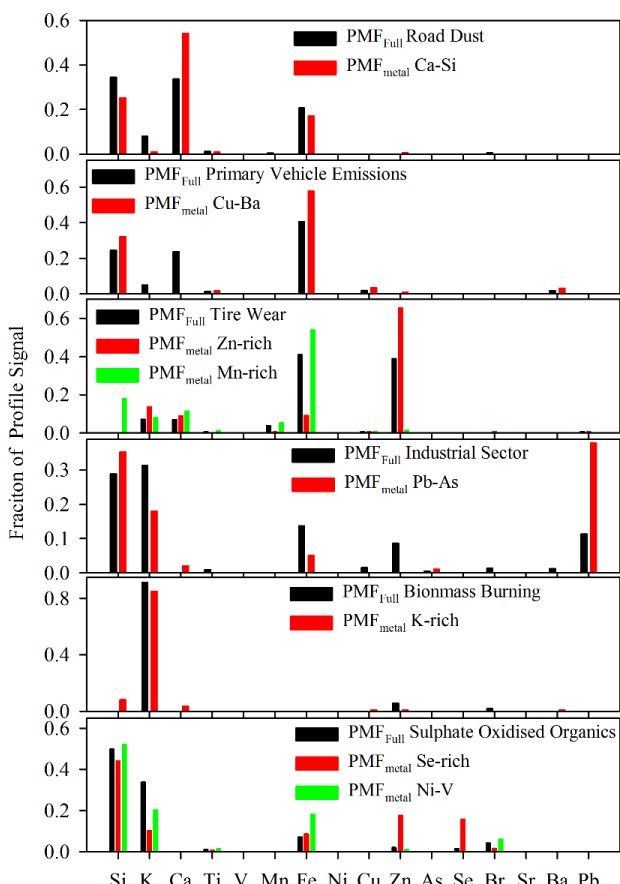

**Figure 6. Comparison of factor profiles from the PMF_Full analysis with comparable factors from the PMF_metal analysis. The bars represent the amount of each species apportioned to the total metal of the factor.**





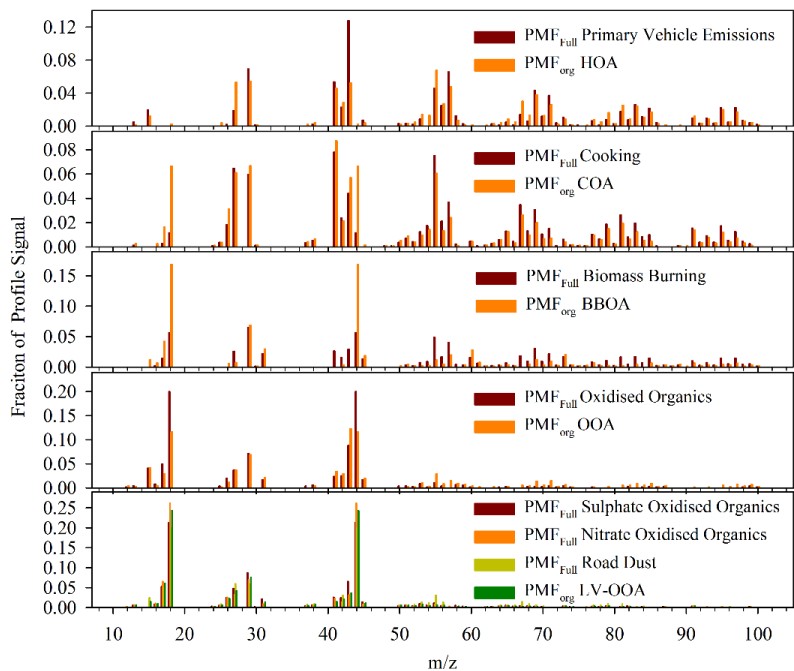

**Figure 7. Comparison of factor profiles from the PMF$_{Full}$ analysis with comparable factors from the PMF$_{org}$ analysis. The bars represent the amount of each mass spectra signal apportioned to the total organic fraction signal of the factor.**





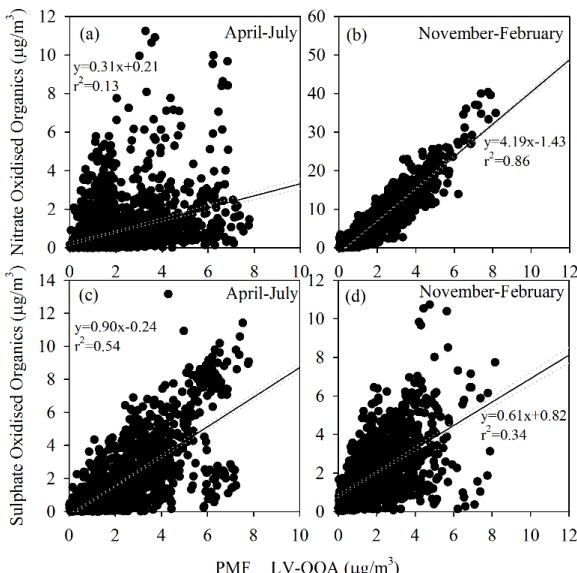

**Figure 8. Scatter plots for PMF$_{org}$ LV-OOA with PMF$_{Full}$ Nitrate and Oxidised Organics (a, b) and PMF$_{Full}$ Sulphate and Oxidised Organics (c, d) comparing their contributions in the warmer months (a, c) and colder months (b, d).**