# Peer review of "Source Apportionment of Urban Particulate Matter using Hourly Resolved Trace Metals, Organics, and Inorganic Aerosol Components"

_Atmospheric Chemistry and Physics, 2016_

## Referee Comment (RC1) · Anonymous Referee #2 · 3 Jun 2016

General comments The paper is focused on source apportionment by Positive Matrix Factorization of hourly resolved particulate matter sampled in Toronto (Canada). The topic is interesting, the data set is large. Obviously the joint use of metal, BC and ACSM data should give better results respect to the use of data from a single instrument; however, the joint use of data coming from different instruments in the PMF analysis it is not trivial and there are many inconsistencies in the reported factor profiles and in the interpretation of the results; furthermore, many important information are missing and the results of the PMF analysis for the identification of the sources is less clear respect to a traditional 24 h based study. Therefore, I cannot suggest the article for publication.

[Figure]

Specific comments: 1) The uncertainty of each metal is normally related to the square root of the number of counts in the peak; it depends on the number of counts in the peak, therefore it is not fix for each element, it varies from spectra to spectra and it cannot be obtained from the measurements on a standard (formula (1) S.I. pag.1); this method clearly underestimate the uncertainty (which is a crucial parameter for PMF analysis) 2) PM2.5 Mass concentrations were measured at the 4th floor rooftop. Nothing is said about the positioning of the other instruments except they are 15 m apart from a main road. At which height? This is a crucial information. If the PM2.5 monitor and the other instruments are not positioned at the same height, the bias in the quantitative PMF results is relevant. 3) In my opinion, the combination of PM1 data for ACSM and PM2.5 data may introduce artefacts in PMF analysis. The reduction of the CE down to 0.5 may be arbitrary. The PM2.5 to PM1 may be compound or day dependent. The authors present the regression for the winter period, but the results could be different in the summer period. 4) The combination of stable markers like the metals and the organics (that maybe unstable at higher temperature) is an open problem in the PMF analysis; this may produce artefacts like the presence of factors which are simply related to the difference in the ratio between some markers in different seasons. The authors do not discuss such a problem that can be the origin of some of the problems in their results 5) Some important information are missing to have an idea of the quality of the PMF results. The graph of the measured PM2.5 mass vs the reconstructed PMF mass must be inserted at least in the SI. Another necessary information is a table with the percentage of the mass of each element or compound explained by the PMF (by all the identified sources) respect to the measured element/compound mass (which is something different from what is reported in table 2). Furthermore, which is the percentage of the mass of each of the identified sources explained by the measured elements/compounds? If all the elements/compounds present in the source are measured this percentage must be 100%, otherwise less (e.g. for the road dust the oxygen from the oxides of Si, Ca, Fe, Ti is not measured) but sometimes PMF overestimates the mass of the source with a percentage above 100%. The sum of

the percentage due to all the sources reported in tables 2 and 3. is 100%. Which is the percentage of the unexplained mass? It is quite strange if PMF reconstruct 100% of the mass. Without all these information, it is impossible to understand how good the PMF results are. PMF is a powerful instrument but it must be used with caution and many checks must be made before giving the results. All these information are missing. Finally, looking at fig. 1 I can see that the error bars for the contribution of all the typical markers are very high; this is unusual; I suppose therefore that the error on the contribution of each source, too, is very high, but it is not reported in any of the tables. 6) The identification of the road dust source is questionable. The error bars for all the characteristic markers are very high. The presence of V (50% of its mass in this source is unusual). The percentage of OA is very high compared to what is normally found in many articles (and higher respect to the primary vehicle emission source). No BC is present in this source at variance with the articles reported as references. The time trend is not similar to that of traffic or of the typical traffic related gases. There is a similarity just at the beginning of the morning, but no decrease after the rush hour and it start decreasing at 3 pm while traffic start decreasing at 6 pm. Finally, it is known that the contribution of the road dust is mainly in the coarse fraction while the exhaust component is mainly in the PM2.5 fraction. It seems quite unreliable that the road dust contributes similar to the exhaust in the PM2.5 fraction or less if we consider also the tire wear source (see below for the problems in the identification of that source). This is probably due to the weakness of the profiles of the identified sources or to the fact that this source is not a traffic related source. In the reference Amato et al (2016) a similar source was identified but it was attributed to a local soil source related to anthropogenic activities, among which construction works. The weekend/weekday ratio would be the same as that of the traffic source. Which are the Enrichment Factors respect to Si of the crustal elements? 7) The identification of the Primary vehicle Emission source is more sounded, looking at its time trend in comparison with that of the traffic related gases or of the traffic counts; however also in this case the profiles shows some problems. Again the errors for the main markers are very high; a comparable contribution

(micrograms/m3) from Si, Ca, Fe and BC is unreasonable in this source, where OC and BC are the main components (see many articles on PMF identification of traffic sources, e.g. the references reported in the article like Amato et al., 2016). This could be due to the combined use of data coming from different instruments, to the quality of the data or to the error estimate. The percentage of OC in this source is also too low (27%). 8) The identification of the Tire Wear source is not clear at all. In fig. 2 there is a maximum common with that of NOx or CO but the time pattern is completely different and the weekday/weekend ratio may be typical also of industrial activities. I would not say that r2 = 0.32, 0.43, 0.17 means "a strong correlation between the Tire Wear factor and traffic related pollutants" (p. 8 l.8). Furthermore, as reported in literature, most of the mass from the tire wear is due to the carbon component, Zn is a typical marker but it is a trace, not a main constituent (and Fe gives even a higher contribution). 9) From what I have reported above, the conclusion (p. 14 l. 7) "the traffic related sources accounting for 17% of PM2.5" is not justified. 10) The Industrial sector source is identified by Pb and As, but the time trend of those two elements shown in fig. S4 is completely different. 11) Which are the metals that help in the identification of the cooking source as reported at p. 9 l. 3? There are plenty of articles that identify the cocking source with only organics. 12) In the Biomass Burning source the absolute contribution of K (a tracer) is higher than that of BC, which is quite strange looking at the literature 13) Why the nitrates source has such a correlation with SO2, which is far higher than that of the sulphate source? 14) All the discussion on the PMF with only metals and ACSM data is not very useful. The quality of the metal data is not very good. Si has background problems with a high noise (and Al is not measured, like Na and Cl), V, Ni, As, Sr are present only in very few cases and also Se is present in less than 30% of the cases. I am not sure they can be considered as good variables (no information on good or weak variables is reported). With only 11 "good" elements, they found 8 factors! In the discussion about the identification of the different sources, there are problems similar to the ones reported for the "full" analysis. The identification of 4 traffic sources is arbitrary. I understand that S was not used in the PMFfull analysis because there is the

ACSM sulphate, but there is no reason for not using it in the PMFmetals analysis. Finally, the authors contradict themselves saying "PMFfull analysis did not always enable better resolution of factors". 15) p 13 l- 29-35 The combination in the PMF of stable markers like K with unstable markers (at higher temperatures) like the organics may be problematic. This could explain the different correlations.

Minor comments: 1) Which program was used to fit the X-Ray spectra? 2) How were the raw aethalometer data were corrected (e.g. which mass absorption coefficient was used?) 3) What does it mean that "only good species" (p 5 l. 13) were taken into account when calculating the Q/Qexp ratio? This is a number that is given by the program considering the results as a whole

---

## Referee Comment (RC2) · Anonymous Referee #3 · 15 Jul 2016

The study investigated the sources of the main elements of the PM2.5 based on hourly resolved particulate matter (PM) speciation during two campaign periods by means of Positive Matrix Factorization (PMF) analysis. Separate PMF analyses were conducted using the trace metal only data (PMFmetal) and organic mass spectra only (PMForg), and compared with the PMFFull results. The results presented here are not so innovative for the scientific community working on air quality studies and of difficult understanding because there are many problems to interpret the results. The PMF analysis needs to be performed with a study based on PM filters sampled every 24 h to identify and apportion the emissive sources present on a given area and to compare also the obtained results even with similar source apportionment studies performed in other

sites. A chemical mass closure is not performed due to the lack of a complete chemical speciation of the PM2.5 on the same filters: the measured elemental concentrations represent a minor fraction of PM. The sum of the estimated source contributions and the simultaneous comparison with total measured PM mass is not possible. The study seems to be quite constrained. The meaning of some paragraphs is unclear and I advise the authors to carefully revise the paper to check for clarity before the resubmission. Therefore, in my opinion, the manuscript is not suitable for publication as current version due to the major deficiencies described above.

Specific comments

2.2 Trace metal measurements Lines 15-20: It could be useful to write what kind of filters are used for this analysis.

3.1 PMF of combined data, PMFFull It could be useful to report the parameters of the good quality of the source apportionment study. The comparison between reconstructed and measured mass is not present, the Bootstrap analysis and the evaluation of the errors regarding the contributions are missing too.

3.1.2 Road dust Lines 5-10: the organic component of PM is considered among the variables input of the PMF analysis in the Figure 1 as m/z ratio and not as measure of OC by thermo-analysis instrument. In my opinion, this evaluation should be better explained; anyway, is not enough to apportion the total PM2.5 mass without this kind of measure. Moreover, the BC concentration is considered in the PMF analysis without a correction for EC measure; is true? Are there any kind of evaluation in this field? I would like to understand the way to apportion in this case the organic component of the PM.

3.1.5 Industrial Sector Lines 25-30. I would like to understand why Pb and As are considered as marker elements of industrial sector; were considered some meteorological parameters as velocity and direction of the wind link to this kind of source?

3.2 Comparison of results for the PMFmetal and PMForg analysis Lines 10-15. For this aspect, the comparability with other V/Ni ratios reported in literature is not shown to distinguish the different sources of Heavy oil combustion due to ship or industrial emissions.

Figure 1. Factor profiles of the nine-factor solution (Road Dust, Primary Vehicle Emissions, Tire Wear, Industrial Sector, Cooking, Biomass Burning, Oxidised Organics, Sulphate and Oxidised Organics, Nitrate and Oxidised Organics) from PMFFull analysis including ACSM organic mass spectra, ACSM inorganics, Xact metals, and Aethalometer black carbon. The plots presented in this way show the chemical profiles but they are not completely clear. I suggest the author to find a more simple way to show the results. Figure 4. I suggest the authors to show the different contribution in percent and absolute terms one next the other one figures.

[Figure]

---

## Author Comment (AC1) · 5 Aug 2016

**Responses to Reviewers**

**Title: Source Apportionment of Urban Particulate Matter using Hourly Resolved Trace Metals, Organics, and Inorganic Aerosol Components**

**Author(s): C.-H. Jeong et al.**

We thank all the referee for providing insightful comments and suggestions. All comments have been carefully considered and addressed. We believe we have improved the manuscript. The detailed responses to specific points are listed below (text in italics shows the reviewer's comments and corresponding responses are shown in blue).

**Reviewer #2**

**General comments**

The paper is focused on source apportionment by Positive Matrix Factorization of hourly resolved particulate matter sampled in Toronto (Canada). The topic is interesting, the data set is large. Obviously the joint use of metal, BC and ACSM data should give better results respect to the use of data from a single instrument; however, the joint use of data coming from different instruments in the PMF analysis it is not trivial and there are many inconsistencies in the reported factor profiles and in the interpretation of the results; furthermore, many important information are missing and the results of the PMF analysis for the identification of the sources is less clear respect to a traditional 24 h based study. Therefore, I cannot suggest the article for publication.

PMF enables the deconvolution of temporal patterns within time series data, hence the findings resolved are inherently related to the temporal characteristics of the input data. Certainly, 24-hr integrated filter data has traditionally been used for PMF analysis, and these data typically have better accuracy due to larger sample loadings and more sensitive analytical instruments. However, use of other data types is becoming more common. For example, PMF is routinely applied to deconvolve high time resolution aerosol mass spectrometry data, as was done in this study. Use of higher time resolution data also enables application of PMF to short-term intensive campaigns (a month or less) where using traditional methods would have resulted in too few filter samples collected; typically, at least 100 samples are required to obtain a reliable PMF solution.

To the best of our knowledge, this is the first study to use organic aerosol (OA), inorganic species (sulphate, nitrate, ammonium), black carbon, and trace metals data measured at high time resolution for source apportionment. One of the advantages of this approach was the resolution that provided more accurate results in terms of the identification, contributions, and locations of sources that last for short time periods (i.e., industrial plume event) and contain strong diurnal trends (i.e., traffic-related emissions). The combined approach was to compare and contrast PMF solutions using OA mass spectra or metals only as these have commonly been used in source apportionment studies. Since no other study has used this combination of high time resolution data, there is no direct point of comparison for the PMF solutions in this study.

We fully agree care must be taking when using data from different instruments. This often arises even with traditional filter-based data where different methods are used to measure different types of analytes (e.g. metals, ions, organic fractions etc.). PMF allows for incorporation of uncertainties so that data from different instruments receives appropriate weighting. Due to the unique combination of data types used in our study, the solution was very carefully investigated to evaluate the uncertainty in the PMF solution and the stability of the PMF solution upon perturbations to its uncertainty matrix. Detailed descriptions for the evaluation of the PMF solution has been discussed in our response and added in the revised manuscript.

**Specific comments**

1) The uncertainty of each metal is normally related to the square root of the number of counts in the peak; it depends on the number of counts in the peak, therefore it is not fix for each element, it varies from spectra to spectra and it cannot be obtained from the measurements on a standard (formula (1) S.I. pag.1); this method clearly underestimate the uncertainty (which is a crucial parameter for PMF analysis)

We agree that the uncertainty estimation is one of the most important steps in positive factorization matrix (PMF) analysis and a comprehensive set of uncertainties is required.

The analytical uncertainty (global or overall uncertainty) of X-ray fluorescence (XRF) analysis is a combination of all sources of random errors and systematic errors introduced by fluctuations of peak and background intensities (i.e., statistical counting errors in the spectra), calibration procedures, matrix effects, and attenuation (e.g., Gutknecht et al., 2010). However, the statistical counting uncertainties based on Poisson statistics were only available for and relevant to the metals measured by the Xact metals monitor. Further, these tended to be small compared to the other sources of error. Alternatively, the overall uncertainty of analytical instruments can be estimated by comparing the replicated results from reference materials (Rousseau 2001). The repeated measurements take into account the errors introduced by the instrument and counting statistics. Thus, the analytical uncertainty in the PMF analysis was empirically estimated through the use of experimental data. Additional uncertainty (i.e., 5%) introduced by the fluctuation of flow rates was also propagated into the overall relative analytical uncertainty in the PMF modeling.

The comparison of the measured statistical counting uncertainties and estimated analytical uncertainty used in the study are depicted in Fig. R1. The statistical counting uncertainties were proportional to the measured concentrations, but were much lower than the analytical uncertainty used in the PMF analysis. By using our approach, the analytical uncertainty used in the PMF analysis encompassed the uncertainty variations from spectra to spectra.

In the Supplement of the revised manuscript, the explanation for the analytical uncertainty of the Xact data has been added.

---

## Author Comment (AC2) · 5 Aug 2016

**Responses to Reviewers**

**Title: Source Apportionment of Urban Particulate Matter using Hourly Resolved Trace Metals, Organics, and Inorganic Aerosol Components**

**Author(s): C.-H. Jeong et al.**

We thank all the referee for providing insightful comments and suggestions. All comments have been carefully considered and addressed. We believe we have improved the manuscript. The detailed responses to specific points are listed below (text in italics shows the reviewer's comments and corresponding responses are shown in blue).

**Reviewer #3**

*The study investigated the sources of the main elements of the PM2.5 based on hourly resolved particulate matter (PM) speciation during two campaign periods by means of Positive Matrix Factorization (PMF) analysis. Separate PMF analyses were conducted using the trace metal only data (PMFmetal) and organic mass spectra only (PMForg), and compared with the PMFFull results.*

*The results presented here are not so innovative for the scientific community working on air quality studies and of difficult understanding because there are many problems to interpret the results. The PMF analysis needs to be performed with a study based on PM filters sampled every 24 h to identify and apportion the emissive sources present on a given area and to compare also the obtained results even with similar source apportionment studies performed in other sites.*

Unlikely other source apportionment studies using 24-hr integrated filter data, hourly data for the chemical species of PM2.5 can be advantageous to identify short-lived sources that have high temporal variations. In addition, PMF is widely used for organic aerosol (OA) mass spectra measured by aerosol mass spectrometer (AMS, Aerodyne) or aerosol chemical speciation monitor (ACSM, Aerodyne) to resolve types of complex OA, one of a large group of compounds in PM2.5. However, without additional information it is not possible to identify the origins of OA in relation to other PM2.5 sources and apportion various OA factions (i.e., low-volatile, semi-volatile organics) to these PM2.5 sources. Metal speciation data have also been used for source apportionment studies (e.g., Dall'Osto et al, 2013). Trace metal based chemical profiles can be very useful for resolving and identifying sources more effectively in the PMF analysis. However, due to the minor contributions of trace metals on total PM2.5 mass (i.e., ~1% of total PM2.5 mass), it is impossible to quantify source contributions of the identified PM2.5 sources using metal data only.

To the best of our knowledge, this is the first reported study to combine OA, inorganic species (sulphate, nitrate, ammonium), black carbon, and trace metal data measured at high time resolution for source apportionment. Furthermore, the effectiveness of the PMF solutions was evaluated in this study by comparing and contrasting three PMF solutions using the combined data, OA only, and metal only. Thus,

we respectfully argue that our methodology and findings are certainly novel and quite innovative. Further the results of this study provide additional insight into PM2.5 sources related to their high temporal variations, various OA fractions, and metal-rich factors. This new knowledge should support the development of control strategies for these sources and thus is of value to the air quality community.

*A chemical mass closure is not performed due to the lack of a complete chemical speciation of the PM2.5 on the same filters: the measured elemental concentrations represent a minor fraction of PM.*

Chemical mass closure was performed as shown in the Supplement (Line 23-30, Page 2, Fig. S2) of the original manuscript. In this study, organics aerosol, sulphate, nitrate, ammonium, black carbon, and trace metals were used in analysis. As shown in Fig. S2, good correlation and agreement were found between the reconstructed mass and total PM2.5 mass measured by a collocated SHARP during the entire period (4234 hourly samples).

Certainly the metals alone represented a minor fraction of the PM which is why these data were combined with the data from the other instruments in the full analysis. No mass closure was performed for the $PMF_{metal}$ for this reason. However, three PMF analyses, $PMF_{Full}$, $PMF_{org}$ and $PMF_{metal}$ were separately conducted and mass closure was done for two out of three of these analyses. These solutions were also compared to evaluate and illustrate the capability of the combined data to identify sources more effectively.

*The sum of the estimated source contributions and the simultaneous comparison with total measured PM mass is not possible.*

As discussed previously, there was good agreement between the sum of the components used in the $PMF_{Full}$ analysis and the measured total PM2.5 mass. Besides, the modelled contributions of $PMF_{Full}$-resolved sources were regressed against the total PM2.5 mass using a multi linear regression method as described in Line 27-29, Page 5. The modelled and measured PM2.5 showed a high correlation ($r^2=0.83$) and agreement with a slope of 0.94. The correlation has been added in the revised manuscript.

*The study seems to be quite constrained. The meaning of some paragraphs is unclear and I advise the authors to carefully revise the paper to check for clarity before the resubmission. Therefore, in my opinion, the manuscript is not suitable for publication as current version due to the major deficiencies described above.*

This is a novel and complex analysis that introduces new concepts requiring more explanation than a traditional PMF paper. We expect that PMF studies with similar combinations of high time resolution data will become more common in the future thus we wanted to help lay down an initial foundation. We have worked to clarify these explanations and interpretations in the revised manuscripts.

*Specific comments*

*2.2 Trace metal measurements Lines 15-20: It could be useful to write what kind of filters are used for this analysis.*

In this study, the continuous metal concentrations were measured by the Xact metals monitor on an hourly basis. The Xact used a Teflon reel tape (manufacturer's recommendation) to sample and analyze simultaneously. In the revised manuscript, the statement has been modified as follows:

"In brief, the Xact instrument pulls ambient air through a section of filter tape (Teflon tape roll) at a flow rate of 16.7 lpm using a PM2.5 sharp cut cyclone (BGI)."

*3.1 PMF of combined data, PMF_{Full} It could be useful to report the parameters of the good quality of the source apportionment study. The comparison between reconstructed and measured mass is not present, the Bootstrap analysis and the evaluation of the errors regarding the contributions are missing too.*

As shown in the caption of Fig. 1, the error bars in the source profiles were estimated by the standard deviations of the 100 bootstrap (BS) runs.

The quality of the 9-factor solution was also discussed in Section 3.1.1. In the revised manuscript, detailed diagnostics of error estimation in the PMF_{Full} solution using the bootstrap and displacement analyses have been added in the Supplement of the revised manuscript. Please refer to the responses for Reviewer 2. The comparison of PMF-modelled PM2.5 mass and measured PM2.5 mass has been added in the revised version as well.

*3.1.2 Road dust Lines 5-10: the organic component of PM is considered among the variables input of the PMF analysis in the Figure 1 as m/z ratio and not as measure of OC by thermo-analysis instrument. In my opinion, this evaluation should be better explained; anyway, is not enough to apportion the total PM2.5 mass without this kind of measure. Moreover, the BC concentration is considered in the PMF analysis without a correction for EC measure; is true? Are there any kind of evaluation in this field? I would like to understand the way to apportion in this case the organic component of the PM.*

Organic carbon (OC) typically determined in 24h filter based PMF studies by thermal/optical techniques (i.e., Sunset Lab OCEC analyzer) is converted to equivalent OA total mass by multiplying by a conversion factor (i.e., 1.4~2.5). This factor accounts for additional mass, mainly hydrogen, associated with organic carbon present in the particle phase. To achieve a proper mass closure, the conversion factor for OC is required and mostly has to be assumed. Thus thermal OC analysis, although very useful, is by no means an absolute or definitive method for determining organic aerosol mass or composition.

In this PMF study we used detailed and more compositionally relevant OA data (as OA mass spectra) measured by the ACSM. With PMF the OA mass spectra can be decomposed into specific groups of m/z

fragments, which are used to identify differences between OA sources (i.e., hydrocarbon-like OA, oxygenated OA, biomass burning OA, low-volatility oxygenated OA). Thus, source apportionment using measured OA fragments can be more effective than the PMF analysis using bulk OC mass to identify sources related to OA. PMF is routinely applied to mass spectra data so as to apportion the organic components in PM (see relevant references within the manuscript).

Due to a different time resolution (1 hr vs. 2 hr interval) and a lower data availability for EC measurements, hourly BC data were used in this study. The optically based BC concentrations measured by the Aethalometer were converted to mass based concentrations using a mass absorption coefficient. Thermally based measurements of EC were not used in the study due to their 2h time resolution but they were available. As shown in Fig. R10, there was good agreement between 2-hour averaged Aethalometer BC and EC measured by a Sunset Lab OCEC analyzer.

[Figure]

Figure R1. Comparison of 2-hr averaged Aethalometer BC (880 nm) with EC measured by a collocated Sunset Lab OCEC analyzer from May 8 to July 2, 2013.

*3.1.5 Industrial Sector Lines 25-30. I would like to understand why Pb and As are considered as marker elements of industrial sector; were considered some meteorological parameters as velocity and direction of the wind link to this kind of source?*

The correlations of the Industrial factor with meteorological parameters (i.e., temperature, RH, wind speed) and the wind sector analysis (i.e., CPF) are shown in Table 2 and Fig. S9 in the original manuscript. Overall, there was no distinct correlation with meteorological parameters. However, the directionality of CPF pointed to the location of a once heavily industrialized sector and a wastewater-treatment facility, indicating the influence of local industrial sources. Recently, Sofowote et al. (2015) also found a similar Pb-As factor associated with non-ferrous metal smelting using 6-year chemical speciation data in Toronto.

*3.2 Comparison of results for the PMFmetal and PMForg analysis Lines 10-15. For this aspect, the comparability with other V/Ni ratios reported in literature is not shown to distinguish the different sources of Heavy oil combustion due to ship or industrial emissions.*

The ratio of V to Ni is generally used as an indicative of ship emissions or residual oil combustion. Viana et al. (2014) reviewed the V/Ni ratios of ship emissions in Europe, which were found that the ratios ranged from 3 to 4. Jeong et al., (2011) reported the V/Ni ratio of 3.7 for ship emissions in Halifax, Canada. Oil combustion from mainly heating for residential and commercial buildings is enriched with Ni and thus the V/Ni ratio for residual oil combustion is typically lower than the ratio for ship emissions (Peltier et al., 2008). In the study, the V/Ni ratio in the $PMF_{metal}$ solution was 1.6, which was much lower than the typical ratio for ship emissions. This may suggest that residual oil combustion for heating purpose is a likely source of the Ni-V factor in this study as Toronto is not a major shipping port.

We have added a statement for the V/Ni ratio in the revised manuscript.

*Figure 1. Factor profiles of the nine-factor solution (Road Dust, Primary Vehicle Emissions, Tire Wear, Industrial Sector, Cooking, Biomass Burning, Oxidised Organics, Sulphate and Oxidised Organics, Nitrate and Oxidised Organics) from PMFFull analysis including ACSM organic mass spectra, ACSM inorganics, Xact metals, and Aethalometer black carbon. The plots presented in this way show the chemical profiles but they are not completely clear. I suggest the author to find a more simple way to show the results.*

We have added a new summary table exhibiting the explained variations of key marker species for each factor in the Supplement of the revised manuscript.

*Figure 4. I suggest the authors to show the different contribution in percent and absolute terms one next the other one figures.*

Thank you for this suggestion. We felt that placing the two figures on top of each other facilitated comparison for each of the factors. We will leave it to the journal to determine if presenting these figures beside each other or on top of each other fits more easily within the manuscript

**References**

Dall'Osto, M., Querol, X., Amato, F., Karanasiou, A., Lucarelli, F., Nava, S., Calzolai, G., and Chiari, M.: Hourly elemental concentrations in PM2.5 aerosols sampled simultaneously at urban background and road site during SAPUSS-diurnal variations and PMF receptor modelling, Atmos. Chem. Phys., 13, 4375-4392, doi:10.5194/acp-13-4375-2013, 2013.

Jeong, C.-H., McGuire, M. L., Herod, D., Dann, T., Dabek-Zlotorzynska, E., Wang, D., Ding, L., Celo, V., Mathieu, D., and Evans, G. J.: Receptor model based identification of PM2.5 sources in Canadian cities, Atmos. Pollut. Res., 2, 158-171, 2011.

Peltier, R. E., Hsu, S.-I., Lall, R., and Lippmann, M.: Residual oil combustion: a major source of airborne nickel in New York City, J. Expos. Sci. Environ. Epidemiol., 19, 603-612, 2008.

Sofowote, U. M., Su, Y., Dabek-Zlotorzynska, E., Rastogi, A. K., Brook, J., and Hopke, P. K.: Sources and temporal variations of constrained PMF factors obtained from multiple-year receptor modeling of ambient PM2.5 data from five speciation sites in Ontario, Canada, Atmos. Environ., 108, 140-150, 2015.

Viana, M., Hammingh, P., Colette, A., Querol, X., Degraeuwe, B., de Vlieger, I., and van Aardenne, J.: Impact of maritime transport emissions on coastal air quality in Europe, Atmos. Environ., 90, 96-105, 2014.